# Comprehensive Benchmark for Tailored Small Molecule-Binding Aptamer Design

## Abstract

Despite their growing role as recognition elements in diagnostics, therapeutics, and biosensing, aptamers remain overlooked by computational design tools compared to antibodies and protein binders. Current pipelines are fragmented and predominantly protein-focused, leaving small-molecule aptamer discovery underexplored. A key bottleneck has been the absence of a unified benchmark dataset that would allow systematic evaluation of predictive and generative models. To address this gap, we introduce the first comprehensive benchmark for aptamer–small molecule interactions, integrating seven curated sources into 2,001 annotated pairs covering 1,309 unique aptamers (DNA and RNA) and 479 chemically diverse ligands. More than half of the entries include quantitative binding affinities, enabling not only binary classification but also regression. To demonstrate the utility of this resource, we establish baseline results across shallow and deep learning baseline models under multiple splitting protocols. Our analysis yields two key insights: (i) the coverage and diversity of aptamer sequences are sufficient to support robust modeling, ensuring that receptor-side representation is not the limiting factor; and (ii) the main challenge arises from the ligand space, where a relatively small number of molecules display high structural diversity, limiting model transferability. Because the ultimate goal is designing aptamers for previously unseen molecules, the observed limitations in ligand transferability point directly to the representation problem, reinforcing the necessity of a common benchmark to address it. By providing a standardized corpus, evaluation protocols, and reproducible baselines, our work establishes a foundation for systematic progress in aptamer–small molecule prediction.

## 1 Introduction

Aptamers are short single-stranded DNA or RNA oligonucleotides capable of binding ions, small molecules, proteins, and even whole cells with high specificity and affinity Keefe et al. (2010); Thiviyanathan & Gorenstein (2012). Their chemical stability, ease of synthesis, and tunability make them attractive alternatives to antibodies in diagnostics, therapeutics, and biosensing. While much of the literature focuses on protein-binding aptamers, small-molecule aptamers are equally critical: they enable detection of metabolites, toxins, drugs, and environmental pollutants—targets that are often inaccessible to antibody-based methods due to their small size or low immunogenicity Ruscito & DeRosa (2016); McKeague & Derosa (2012). Expanding aptamer discovery to small molecules is therefore not merely an extension but a necessary step to broaden the biomedical and chemical applications of this technology.

Experimentally, aptamer selection is most often performed through SELEX (Systematic Evolution of Ligands by Exponential enrichment), an iterative process of binding, separation, and amplification Ellington & Szostak (1990). Despite its impact, SELEX is labor- and time-intensive, and suffers from amplification biases such as PCR drift, which can eliminate rare but functional sequences Komarova & Kuznetsov (2019); Wang et al. (2019). These challenges are amplified in the case of small molecules, where weaker affinities and subtle interactions reduce efficiency and reproducibility Hu et al. (2025); Lam et al. (2022).

In contrast to classical drug discovery, where the target is fixed and receptors are engineered or optimized, aptamer design reverses the task: given a molecular target, one must discover a binding

oligonucleotide. This inversion vastly enlarges the search space of possible candidates. Structure-based methods such as molecular docking are widely used in virtual screening but face scalability limits when applied to large combinatorial libraries Cheng et al. (2012); Zhao (2024). Molecular dynamics simulations provide valuable atomistic insights into flexibility and binding pathways, but their computational cost and limited timescales make them impractical for high-throughput discovery De Vivo et al. (2016); Díaz-Fernández et al. (2025).

Machine learning provides a third path. Predictive modeling has been applied primarily to aptamer–protein systems, where classifiers Patel et al. (2024) and, more recently, generative models Wang et al. (2024); Iwano et al. (2022) have shown promise. However, regression-based predictors of binding strength are still rare, existing classifiers often generalize poorly beyond their training domains, and generative approaches typically depend on large SELEX-derived datasets that are not available for small-molecule targets. As a result, small-molecule aptamers remain largely absent from the current machine learning literature.

Because generalization to unseen molecules is essential, two computational tasks are especially critical. First, **classification** of binding vs. non-binding pairs provides the foundation for virtual screening. Second, **regression** of quantitative affinities supplies continuous signals for optimization and reinforcement learning Bashir et al. (2021). In addition, techniques such as negative sampling and data augmentation are widely used in molecular ML to address class imbalance and improve model robustness Li et al. (2025); Zhang et al. (2024), yet they remain unexplored in the aptamer–small molecule domain due to the lack of standardized datasets. Ultimately, accurate classification and regression are not endpoints but prerequisites for the final goal: **generative design of aptamers** tailored to new molecular targets.

In this work, we present a comprehensive benchmark for aptamer–small molecule interactions, unifying seven curated sources into 2,001 annotated pairs covering 1,309 unique aptamers and 479 ligands. Approximately half of the entries include quantitative binding constants, enabling both classification and regression tasks. We establish baseline results using classical machine learning methods and one representative neural model under multiple evaluation protocols. Our analysis of representation choices, negative sampling strategies, and generalization behaviors highlights current bottlenecks, particularly the diversity of the ligand space. While not intended as an exhaustive study of deep learning methods, this work provides a robust foundation and a standardized dataset for the community, enabling systematic progress in modeling, augmentation, and ultimately the generative design of aptamers for novel small molecules.

## 2 RELATED WORKS

The transformative role of curated benchmarks in machine learning is well established. Resources such as MoleculeNet for small molecules Wu et al. (2017), TAPE for proteins Rao et al. (2019), and FLIP for protein fitness landscapes Dallago et al. (2022) have shown how standardized datasets and protocols can catalyze progress by enabling fair comparisons, reproducibility, and rapid iteration.

In the case of aptamers, the situation is less mature. Early efforts to model aptamer–protein interactions date back to Li et al. (2014), who combined pseudo–amino acid composition features with Random Forest classifiers, achieving modest accuracy (MCC $\approx 0.37$) Li et al. (2014). Follow-up studies extended this dataset with ensemble learning methods Zhang et al. (2016) and more recently transformer-based architectures such as AptaTrans Shin et al. (2023). Generative approaches including AptaGPT Ding et al. (2024), RaptGen Iwano et al. (2022), and AptaDiff Wang et al. (2024) have leveraged SELEX data to produce candidate aptamer sequences, but these rely on large RNA–protein datasets that are rarely available and offer zero coverage for small-molecule systems. Importantly, even within aptamer–protein studies, regression-based predictors of quantitative affinity remain scarce, limiting their usefulness for optimization or generative design pipelines.

Structure-based methods provide another angle. Models such as AiDTA Guo et al. (2025) and geometry-conditioned frameworks like RhoDesign Wong et al. (2024) attempt to guide sequence generation through molecular docking or shape-conditioned learning. While these techniques offer mechanistic insights, they remain heavily protein-centric, dependent on well-characterized complexes, and difficult to scale to the chemically diverse space of small molecules. Moreover, dock-

ing and molecular dynamics simulations, though powerful, remain computationally prohibitive for large-scale aptamer screening.

Attempts to move toward small-molecule aptamer prediction exist, but remain limited. Smart-SELEX Douaki et al. (2022) and AptaBERT Morsch et al. (2023) indicate interest in this direction, yet neither provides open datasets or fully documented models. As a result, reproducibility and reuse are hindered, making it difficult for the community to build upon prior work. Beyond individual modeling approaches, several databases of aptamer interactions have been released, including RSAPred Krishnan et al. (2024), AptamerBase, Apta-Index (AptaGen), UTexas Askari et al. (2024), RiboCentre, and AptaDB Chen et al. (2024), alongside more recent manual curation efforts (see Appendix). While each provides valuable information, they differ markedly in scope: some focus on RNA–protein systems, others emphasize DNA aptamers, and coverage of small molecules is often sparse or inconsistent. Annotation practices also vary, with quantitative binding constants available only for subsets of entries. As a result, these datasets remain fragmented, limiting their utility for training generalizable models and preventing consistent evaluation across studies.

Taken together, current approaches have established useful foundations, but they remain fragmented: classification models often fail to generalize beyond their training domain, generative models depend on scarce SELEX data, and structure-based pipelines are computationally costly and protein-biased. Crucially, no existing resource provides a unified, openly accessible benchmark for small-molecule aptamer interactions. This absence prevents systematic evaluation of representation strategies, negative sampling protocols, or augmentation methods. Our work addresses this critical gap by introducing a curated, standardized benchmark dataset accompanied by reproducible baselines, aiming to bring the rigor of benchmark-driven development to aptamer–small molecule modeling.

## 3 BENCHMARK DATASET

### 3.1 SOURCES AND INTEGRATION.

Our benchmark integrates data from seven curated sources: RSAPred, AptamerBase, Apta-Index (AptaGen), UTexas, RiboCentre, AptaDB, and a manually curated collection described in the Appendix (Table 4, Table 5). Each of these databases was developed with a distinct focus: RSAPred emphasizes RNA aptamers with quantitative affinity data, AptamerBase is DNA-dominant but only partially annotated, AptaGen is small but dense in $K_d$ labels, UTexas provides long diverse sequences, RiboCentre contains RNA-only entries, and AptaDB adds DNA-heavy positives without regression data. To complement these heterogeneous sources, our manually curated subset from fills critical gaps by providing a larger share of negative examples, ligands of higher biomedical relevance (e.g., toxins and drugs), and sequence-level variants such as single-nucleotide mutations. This balances the strong bias toward positives and protein-oriented targets in existing databases, making the combined resource more representative for small-molecule aptamer modeling.

### 3.2 DATASET STATISTICS.

After cleaning and deduplication, the merged dataset contains **2,001 aptamer–molecule pairs**, covering **1,309 unique aptamer sequences** and **479 unique small molecules**. Approximately 58% of entries (1,161 pairs) include dissociation constants ($K_d$), enabling both classification and regression tasks. The dataset remains skewed toward positive binders (1,842 vs. 159 negatives), reflecting common reporting biases in experimental studies. Aptamer sequences range from very short ($< 20$ nt) to long ($> 100$ nt), with an average length of $48.9 \pm 26.7$ nucleotides. Ligands span a broad chemical space, with molecular weights ranging from small metabolites ($< 200$ Da) to large drug-like compounds ($> 1000$ Da), with a mean of $534.8 \pm 677.4$ Da. The overall logP distribution is centered near zero ($-0.44 \pm 4.73$), covering both polar and hydrophobic molecules. Figure 1 summarizes key distributional properties of the dataset, including aptamer sequence lengths, pairwise sequence diversity (Levenshtein distances), chemical diversity of ligands (Tanimoto similarity), and the range of experimental $pK_d$ values. These distributions highlight that while sequence coverage is broad and sufficiently diverse to support robust modeling, the ligand space is comparatively sparse yet structurally heterogeneous, making generalization across new diverse molecules the primary bottleneck for predictive methods.

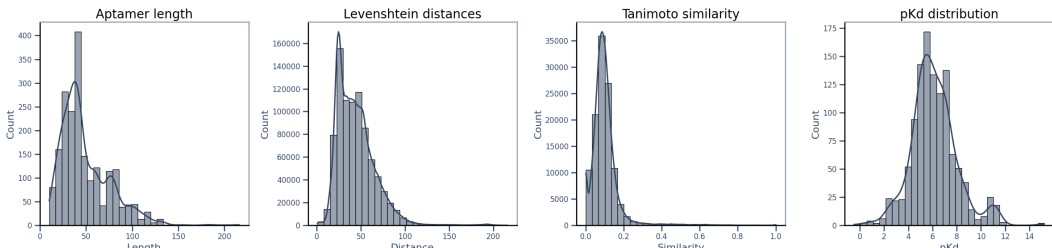

Figure 1: Distributional properties of the benchmark dataset. (Left to right) (i) Aptamer sequence lengths, (ii) pairwise Levenshtein distances between sequences, (iii) Tanimoto similarity between molecular fingerprints, and (iv) distribution of experimental $pK_d$ values. Together these plots characterize sequence and ligand diversity as well as the dynamic range of affinity measurements.

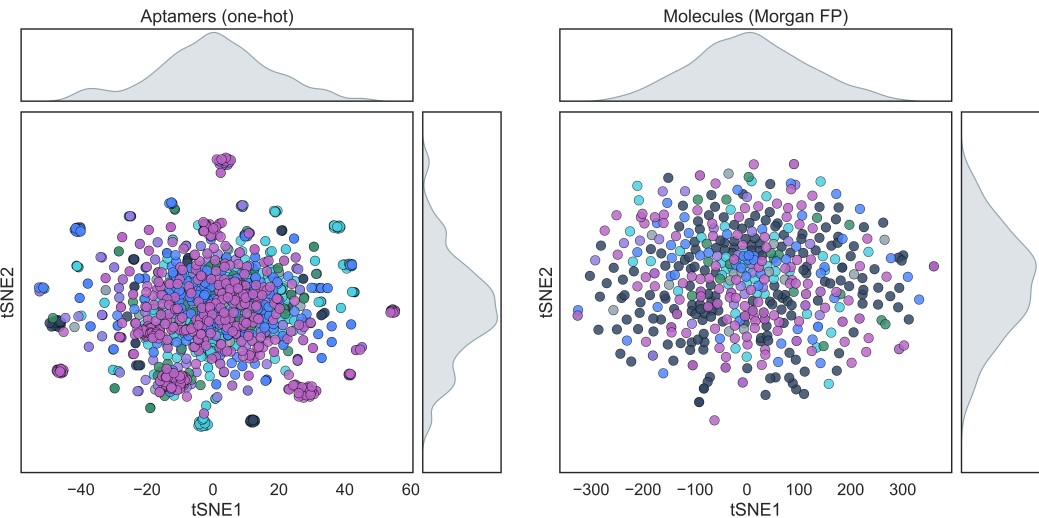

Figure 2: t-SNE projections of aptamer sequences (left) and target molecules (right), colored by origin.

## 3.3 APTAMER AND LIGAND SPACE

To further characterize the dataset, we visualized aptamer sequences and ligands in two-dimensional t-SNE projections (Figure 2). Aptamers were encoded using one-hot nucleotide representations, while molecules were represented with Morgan fingerprints. Each point corresponds to an individual sequence or ligand, colored by its dataset of origin.

For aptamers (left panel), the projections reveal clusters of closely related sequences, particularly within RSAPred and AptaGen, reflecting redundancy and family-like motifs within these sources. In contrast, Manual and AptaDB contribute a broader, more diffuse distribution, adding diversity that complements the more redundant subsets. This suggests that while the overall sequence space is well covered, some datasets provide highly overlapping variants, whereas others expand the landscape with more distinct sequences.

For ligands (right panel), the distribution is more dispersed and lacks tight clustering, indicating high chemical heterogeneity. Compounds from different sources largely overlap in the same chemical space, spanning drug-like molecules, metabolites, and small organic compounds. This structural diversity, coupled with the relatively small number of unique molecules, makes the evaluation of molecular representations particularly critical in downstream tasks.

### 3.4 ANNOTATION COMPLETENESS.

Annotation quality is uneven across the benchmark: while 1,161 out of 2,001 pairs (58%) include quantitative $K_d$ values, the remainder are annotated only with binary activity labels. At the same time, the binary classification task is strongly imbalanced, with positives constituting 92% (1,842 pairs) and negatives only 8% (159 pairs) of the corpus (Appendix, Table 4). This skew reflects typical experimental reporting practices and poses challenges for training robust classifiers, reinforcing the need for systematic negative sampling and augmentation strategies.

### 3.5 DATA SPLITS.

To support reproducible evaluation, we define three complementary splitting protocols: (i) **stratified group splits** preserving label balance across folds, (ii) **aptamer-disjoint splits**, and (iii) **molecule-disjoint splits**.

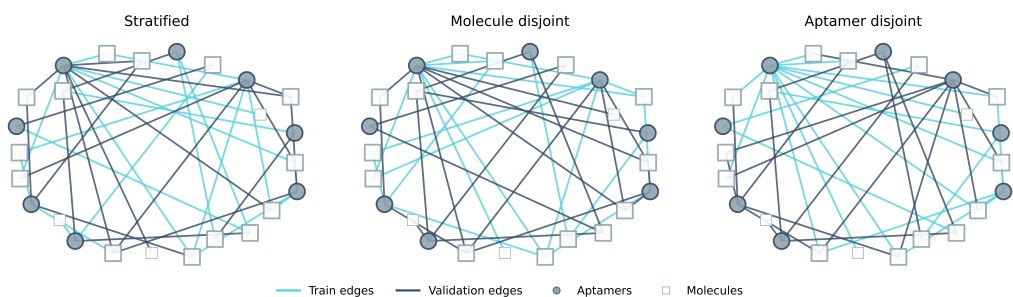

Figure 3: Illustration of the three data splitting strategies (stratified, aptamer-disjoint, and molecule-disjoint) represented as graphs of interactions between aptamers and molecules.

In stratified group splits (Figure 3, left), individual aptamer–molecule pairs are assigned to training or test sets such that the overall distribution of binding affinities remains balanced. Importantly, this scheme does not enforce exclusivity over entities: the same aptamer or the same molecule may occur in both training and test sets, albeit in association with different partners. For example, an aptamer tested against two ligands may contribute one pair to training and the other to testing.

In contrast, in molecule-disjoint splits (Figure 3, center), exclusivity is applied to ligands rather than sequences. All binding interactions of a given molecule are confined to one partition, such that the model is forced to generalize across unseen ligands even if the training set contains overlapping aptamers. This configuration highlights the ability of the model to predict binding outcomes for entirely new molecular scaffolds..

Finally, aptamer-disjoint splits (Figure 3, right) group all ligand interactions associated with a given sequence together. If an aptamer binds to multiple molecules, all resulting pairs are assigned either entirely to training or entirely to testing. This ensures that no sequence representation seen during training reappears in validation or test evaluation, better reflecting the challenge of predicting binding for completely novel aptamers.

These distinct splitting strategies correspond to different practical scenarios. Stratified splits simulate performance in settings where both aptamers and molecules are partially known but new interactions are being discovered. Aptamer-disjoint splits address the challenge of screening entirely novel sequences, such as in de novo aptamer design. Molecule-disjoint splits evaluate generalization to new ligands, relevant for applications in biosensors design where predicting binding for previously uncharacterized molecules is critical. Together, these approaches provide a comprehensive framework for assessing model robustness across varied real-world tasks.

## 3.6 EXPERIMENTS

### 3.6.1 IMPLEMENTATION DETAILS

We evaluate models under three evaluation protocols. Negative samples were generated by cross-pairing aptamers and small molecules from the dataset for which no confirmed interactions are known. For each positive interaction, $n$ negative pairs were created by pairing the aptamer with $n$ different small molecules, effectively multiplying the dataset size by a factor of $n$. This approach, previously applied in protein–aptamer interaction studies, balances the training data by mixing entities to create presumed non-binding pairs Li et al. (2014). However, it does not guarantee true absence of interaction, as some cross-paired samples may still bind but remain uncharacterized.

**Aptamers and Molecules encoding**  Aptamers were encoded as $k$-mers, one-hot encodings, or pretrained oligonucleotide embeddings Fishman et al. (2025), while ligands were encoded by fingerprints, descriptors, or ChemBERTa embeddings Chithrananda et al. (2020) depending on the setup. All shallow and deep learning models were compared with tuned hyperparameters, reporting ROC-AUC and MCC for classification, and RMSE, MAE, $R^2$, $r_p$, and $r_s$ for regression.

**Model training**  We evaluated LightGBM classifier. Feature selection was performed using LightGBM gain, and hyperparameters (e.g., num_leaves, learning rate, subsample) were tuned with Optuna on training folds.

**Deep Learning Model Architecture**  The baseline deep learning model employs pretrained GENA-LM and ChemBERTa transformers to encode aptamer sequences and molecular SMILES, respectively. On top of frozen or partially fine-tuned embeddings, several architectural variants were evaluated, namely, *Identity* (direct projection with optional linear mapping), *CNN* (1D convolutions with global pooling), *LSTM* (bidirectional LSTM with attention pooling), and *Transformer* blocks (see Appendix for details). The aptamer and molecule embeddings are concatenated and fused via a linear projection with normalization and dropout, then classified by a single-hidden-layer multilayer perceptron (MLP) producing binding logits.

**Training protocol**  Training fine-tunes the last two layers of each encoder; the remaining layers are frozen. Detailed training settings such as epochs, learning rate, optimizer, and other parameters are provided in the Appendix C.5.

### 3.6.2 BASELINE ML MODELS

We compare shallow models (LGBM, MLP, RF); results of model screening are presented in Appendix B. Given strong data imbalance (159 negatives vs. 1842 positives), we apply synthetic negative sampling and evaluate primarily with ROC-AUC and MCC. A sanity check with randomized features shows that PR/F1 remain inflated under imbalance, while ROC-AUC and MCC collapse to chance. A sweep over augmentation ratios highlights 1:3 negatives-to-positives as optimal.

To probe generalization, we enforce identity-disjoint splits. Results for the best LGBM configuration ($k$-mer(4) + ChemBERTa) are shown in Table 1. Performance drops moderately when holding out new aptamers, but sharply when holding out new molecules, suggesting that pooled molecular embeddings are less transferable to unseen chemical structures. In this case, Morgan fingerprints can match or even outperform ChemBERTa on MCC, highlighting that simple descriptors remain competitive when cross-modal interactions are not modeled.

For ranked screening, ROC-AUC captures the quality of orderings across thresholds, while MCC directly reflects thresholded utility under imbalance. With feature selection and parameter optimization we lift MCC to $0.70 \pm 0.02$ on grouped CV, with ROC-AUC $\approx 0.91$. Feature analysis showed that aptamer $k$-mer and ChemBERTa features contribute nearly equally to the LGBM decision function (59% vs. 41%), confirming that both feature spaces provide complementary and non-redundant information for classification.

Having established robust tabular baselines, we next compare a set of deep learning architectures to assess whether end-to-end training on pretrained encoders can better capture cross-entity interaction patterns.

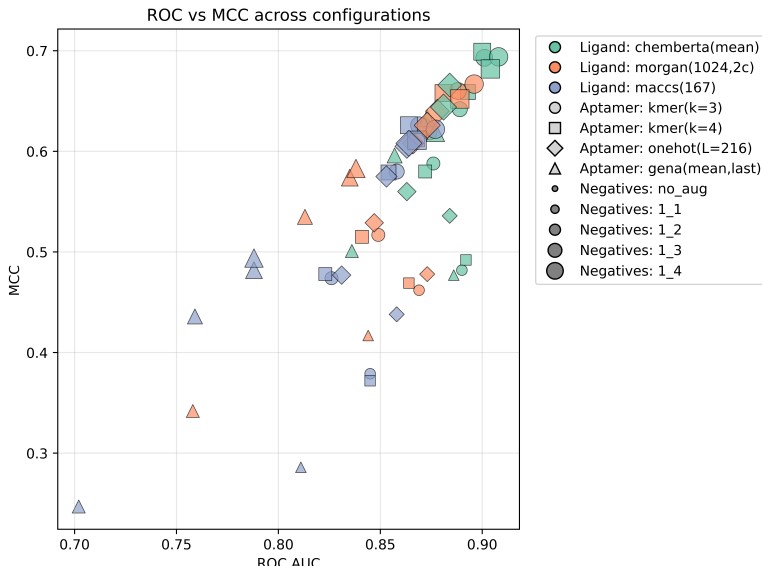

Figure 4: ROC-AUC vs. MCC across descriptor combinations and negative sampling ratios for LGBM. Colors indicate ligand encodings, shapes indicate aptamer encodings, and marker size denotes the number of negatives per positive. $k$-mer features with ChemBERTa embeddings dominate the upper-right region, while negative-to-positive ratios of 1:3–1:4 maximize MCC.

Table 1: Sanity check and identity-disjoint generalization for LGBM with aptamer $k$-mer(4) on the original dataset (no negative sampling). Metrics are mean±std over 5 folds.

| Split protocol | Molecule representation | ROC-AUC ↑ | MCC ↑ |
|---|---|---|---|
| Sanity check (random features) | — | $0.451 \pm 0.020$ | $0.000 \pm 0.000$ |
| Stratification by label | ChemBERTa | $0.892 \pm 0.009$ | $0.673 \pm 0.019$ |
| Aptamer-identity disjoint | ChemBERTa | $0.878 \pm 0.020$ | $0.630 \pm 0.028$ |
| Molecule-identity disjoint | ChemBERTa | $0.660 \pm 0.019$ | $0.342 \pm 0.041$ |
| Molecule-identity disjoint | Morgan (1024, $r$=2) | $0.663 \pm 0.022$ | $0.384 \pm 0.060$ |

### 3.6.3 DL BASELINE MODELS

As shown in the previous section, tabular baselines based on $k$-mer and molecular fingerprints with LGBM already achieve strong and stable performance, but they rely on shallow concatenation and struggle with molecule-disjoint generalization. In related aptamer-protein studies, pretrained encoders combined with lightweight architectural heads have demonstrated stronger cross-entity transfer Patel et al. (2024). Motivated by these observations, we extend the same scheme to aptamer-small molecule prediction, systematically screening multiple deep learning designs.

Table 2 summarizes the top-performing configurations. Under stratified group splits, the three best models (Identity-LSTM, Identity-Transformer, Identity-CNN) achieve nearly identical performance (MCC $\approx 0.41$), suggesting that once pretrained embeddings are available, the specific choice of top-layer encoder has only a secondary effect. All strong models relied on partial unfreezing of the last two layers consistently yielded higher stability than full freeze as shown in Appendix C.6, Table 7.

Despite this consistency, deep models underperform compared to the tabular LGBM baseline (MCC $\approx 0.70$), highlighting that basic end-to-end pre-trained DL does not yet close the gap. Performance drops moderately under aptamer-disjoint splits (MCC $\approx 0.41$), showing that GENA-LM representations transfer reasonably well to unseen sequences. By contrast, molecule-disjoint splits

expose a severe weakness: MCC falls to $0.18$, underscoring the limited generalization capacity of ChemBERTa embeddings for unseen ligands.

Table 2: Top DL configurations across split types. Metrics are mean±std over 5 folds. All use gated fusion; aptamer encoder = GENA-LM, molecule encoder = ChemBERTa.

| Split | Model | Partial Unfreeze | MCC | ROC-AUC |
|---|---|---|---|---|
| Grouped | Identity–LSTM | Last 2 layers | $0.412 \pm 0.034$ | $0.726 \pm 0.028$ |
| Grouped | Identity–Transformer | Last 2 layers | $0.413 \pm 0.029$ | $0.722 \pm 0.029$ |
| Grouped | Identity–CNN | Last 2 layers | $0.408 \pm 0.028$ | $0.731 \pm 0.021$ |
| Aptamer-disjoint | Identity–LSTM | Last 2 layers | $0.407 \pm 0.025$ | $0.721 \pm 0.023$ |
| Molecule-disjoint | Identity–LSTM | Last 2 layers | $0.186 \pm 0.051$ | $0.585 \pm 0.040$ |

In summary, this study shows that deep learning architectures, even when equipped with strong pretrained encoders, remain weaker than optimized LGBM baselines. The main limitation arises from molecular embeddings, which fail to generalize to unseen ligands, whereas aptamer representations from GENA-LM transfer more reliably to novel sequences. Within this regime, the choice of top-layer architecture exerts only a minor effect: CNN, LSTM, and Transformer heads all converge to similar performance. Partial unfreezing of molecule encoder layers was essential to achieve competitive results. Taken together, these findings emphasize that in realistic discovery settings where diverse chemical structures dominate, the quality of ligand representations, rather than model architecture, dictates predictive success. Interestingly, a similar pattern was recently reported for aptamer–protein prediction: the APIPred modelFang et al. (2024), based on XGBoost and handcrafted features, outperformed deep learning counterparts.

### 3.6.4 REGRESSION TASK

Since the combination of $k$-mer(4) descriptors for aptamers and Morgan fingerprints for ligands proved to be the strongest setup in classification (for the most challenging molecule-disjoint split), we adopted the same representation for regression to predict quantitative binding affinities. Using an Optuna-tuned LightGBM regressor, we obtained the following cross-validation results:

Table 3: Cross-validation metrics for LGBMRegressor with $k$-mer(4) + Morgan fingerprints (mean ± std over folds).

| Model | RMSE ↓ | MAE ↓ | $R^2$ ↑ | $r_P$ ↑ | $r_S$ ↑ |
|---|---|---|---|---|---|
| LGBMRegressor | $2.42 \pm 0.13$ | $1.51 \pm 0.10$ | $0.458 \pm 0.034$ | $0.678 \pm 0.025$ | $0.624 \pm 0.029$ |

Feature importance analysis revealed that aptamer descriptors contributed the majority of predictive signal ($71\%$ of total gain), with molecular fingerprints accounting for the remaining $29\%$. Top-ranked features included specific $k$-mer indices and several Morgan bits, indicating that the model captures a mix of sequence-level motifs and chemical substructures. While errors remain non-negligible, the observed correlations confirm that regression models can prioritize candidates by predicted binding strength rather than binary outcome, thus extending their utility for ranking and filtering in real-world aptamer screening pipelines.

### 3.6.5 FUTURE DIRECTIONS

Our results highlight two primary challenges. First, the molecular dataset contains a relatively small number of unique ligands (e.g. comparing to drug discovery tasks), yet these represent a maximally diverse chemical space. This structural diversity makes it difficult for current embeddings to generalize well to unseen molecules. Future work will explore more expressive representations (e.g., graph neural network encoders) that better capture detailed chemical structure. Second, our current gated fusion approach is limited; more sophisticated cross-entity integrations, such as cross-attention layers, may better model sequence-ligand interactions. Finally, planned integration of classification

and regression models into a reinforcement learning framework aims to guide aptamer sequence generation toward higher binding affinity, forming a complete generative design pipeline.

# 4    CONCLUSION

In this work, we presented a comprehensive benchmark for aptamer–small molecule recognition, combining seven curated datasets into a unified corpus with classification and regression labels. Through systematic evaluation of shallow and deep learning models, we found that simple tabular approaches (LightGBM with $k$-mer and fingerprint features) currently outperform end-to-end deep learning architectures. This performance gap largely stems from the challenge of generalizing across a small but structurally diverse set of unique molecules.

The use of rigorous data splitting protocols, including molecule-identity disjoint splits, underscores the difficulty in predicting binding for novel chemical scaffolds. Although top-layer architectural variations only slightly affect performance, enhancing ligand representations and cross-modal fusion remains crucial.

In summary, this benchmark offers a standardized evaluation framework, clarifies existing limitations in aptamer–ligand prediction, and points toward data-driven generative modeling beyond traditional approaches such as SELEX and docking.

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

# A APPENDIX

## A.1 DATASET DESCRIPTIONS

**RSAPred.** RSAPred contains 513 aptamer–molecule pairs, strongly dominated by RNA sequences (22 DNA vs. 491 RNA). It includes 139 unique aptamers and 195 unique small molecules, with relatively short sequences on average ($33.1 \pm 21.7$ nt). All entries are annotated with quantitative binding constants, making this dataset particularly valuable for regression tasks. Its molecules are of moderate size ($517.3 \pm 227.1$ Da) and activity labels are relatively balanced (455 positives vs. 58 negatives). The average logP is slightly negative ($-0.71 \pm 4.32$), reflecting predominance of polar ligands.

**AptamerBase.** AptamerBase comprises 334 entries, with a stronger DNA representation (205 DNA vs. 129 RNA). It covers 312 unique aptamers and 52 molecules, with medium-length sequences ($55.3 \pm 24.7$ nt). About 44% of entries (148) include $pK_d$ annotations, making it partially suitable for regression. The ligands tend to be heavy and highly variable in size ($779.7 \pm 1158.0$ Da), with a more hydrophilic character ($-1.63 \pm 6.64$ logP). Labels are skewed toward positives (323 vs. 11 negatives).

**AptaGen.** AptaGen is the smallest dataset, with only 36 entries and 35 unique aptamers. Despite its size, it provides dense quantitative annotation, with $pK_d$ values available for 30 entries. The sequences are short ($38.8 \pm 17.8$ nt), while the 27 ligands exhibit extreme variability in molecular weight ($832.6 \pm 1390.6$ Da). Its activity distribution is heavily positive (33 vs. 3 negatives), limiting classification utility but supporting regression benchmarking. The ligands show strongly negative mean logP ($-1.84 \pm 5.53$), indicating polar bias.

**UTexas.** UTexas contains 122 entries and 117 unique aptamers, showing strong sequence diversity. It is moderately balanced between DNA and RNA (76 vs. 46) and features long sequences ($80.4 \pm 26.4$ nt). The dataset includes 47 unique molecules of intermediate weight ($767.8 \pm 974.5$ Da) and more hydrophobic character ($1.57 \pm 7.35$ logP). Most entries (103) include $pK_d$ values. Its activity distribution is skewed toward positives (111 vs. 11 negatives).

**RiboCentre.** RiboCentre consists of 85 RNA-only entries, covering 84 unique aptamers and 37 distinct molecules. It contains some of the longest sequences in the benchmark ($77.1 \pm 37.6$ nt), while ligands are of moderate size ($479.8 \pm 470.4$ Da). No quantitative binding constants are available, restricting its use to classification. All entries are labeled positive, making it unsuitable for training balanced classifiers. The mean logP is $-0.30 \pm 4.49$.

**AptaDB.** AptaDB includes 257 entries and contributes 230 unique aptamers and 88 ligands, with DNA dominating (192 DNA vs. 65 RNA). Its sequences are moderately long ($58.4 \pm 22.8$ nt), while ligands are chemically diverse ($531.2 \pm 798.9$ Da) and moderately polar ($-0.30 \pm 3.80$ logP). No $pK_d$ annotations are provided, and all records are labeled positive, restricting the dataset to positive-only classification tasks.

Table 4: Overview of curated aptamer–small molecule datasets. For each dataset we report: number of rows, DNA:RNA ratio, number of unique aptamers, mean aptamer length with standard deviation (len [nt]), number of unique target molecules, mean target molecular weight with standard deviation (MW [Da]), mean logP, number of entries with quantitative binding constants (with pKd), and active vs inactive counts (pos:neg). Combined statistics are computed after deduplication and integration.

| dataset | rows | DNA:RNA | uniq apt | len [nt] | uniq SM | MW [Da] | logP | with pKd | pos:neg |
|---|---|---|---|---|---|---|---|---|---|
| RSAPred | 513 | 22:491 | 139 | $33.1 \pm 21.7$ | 195 | $517.3 \pm 227.1$ | $-0.71 \pm 4.32$ | 513 | 455:58 |
| AptamerBase | 334 | 205:129 | 312 | $55.3 \pm 24.7$ | 52 | $779.7 \pm 1158.0$ | $-1.63 \pm 6.64$ | 148 | 323:11 |
| AptaGen | 36 | 19:17 | 35 | $38.8 \pm 17.8$ | 27 | $832.6 \pm 1390.6$ | $-1.84 \pm 5.53$ | 30 | 33:3 |
| UTexas | 122 | 76:46 | 117 | $80.4 \pm 26.4$ | 47 | $767.8 \pm 974.5$ | $1.57 \pm 7.35$ | 103 | 111:11 |
| Ribocentre | 85 | 0:85 | 84 | $77.1 \pm 37.6$ | 37 | $479.8 \pm 470.4$ | $-0.30 \pm 4.49$ | 0 | 85:0 |
| AptaDB | 257 | 192:65 | 230 | $58.4 \pm 22.8$ | 88 | $531.2 \pm 798.9$ | $-0.30 \pm 3.80$ | 0 | 257:0 |
| Manual | 654 | 516:138 | 515 | $45.2 \pm 20.4$ | 157 | $372.2 \pm 223.8$ | $0.02 \pm 3.10$ | 367 | 578:76 |
| **Combined** | **2001** | **1030:971** | **1309** | $\mathbf{48.9 \pm 26.7}$ | **479** | $\mathbf{534.8 \pm 677.4}$ | $\mathbf{-0.44 \pm 4.73}$ | **1161** | **1842:159** |

**Manual Curation.** The manually curated dataset is the largest single source, with 654 entries. It contains 515 unique aptamers and 157 distinct molecules, with shorter sequences than other datasets ($45.2 \pm 20.4$ nt). Its ligands are lighter ($372.2 \pm 223.8$ Da) and chemically diverse, with nearly neutral mean logP ($0.02 \pm 3.10$). A substantial fraction (367) include $pK_d$ annotations, and the activity distribution is more balanced than in other datasets (578 positives vs. 76 negatives). This subset was designed to contribute additional negatives, more biomedically relevant small molecules (e.g. drugs, toxins), and single-nucleotide variants, complementing the positive-heavy and protein-oriented bias of other resources.

**Combined Dataset.** Merging all sources yields 2,001 entries covering 1,309 unique aptamers and 479 unique ligands after deduplication and integration. The average sequence length is $48.9 \pm 26.7$ nt, and molecules span a broad chemical space with high variance ($534.8 \pm 677.4$ Da). The overall mean logP is $-0.44 \pm 4.73$, reflecting a mixture of polar and hydrophobic compounds. In total, 1,161 entries contain quantitative binding constants, while binary activity is annotated for all (1,842 positives vs. 159 negatives). This combined dataset forms the basis of our benchmark, providing both breadth and diversity for systematic evaluation of aptamer–small molecule modeling.

Table 5: Pairwise overlap between datasets (all entries included). Each cell shows "entries / unique aptamers / unique small molecules". Diagonal entries correspond to dataset totals.

| | RSAPred | AptamerBase | AptaGen | UTexas | RiboCentre | AptaDB | Manual |
|---|---|---|---|---|---|---|---|
| RSAPred | 487/139/195 | 0/0/4 | 0/1/5 | 0/0/5 | 2/4/12 | 0/1/7 | 8/2/19 |
| AptamerBase | 0/0/4 | 312/312/52 | 2/2/6 | 6/7/15 | 1/1/7 | 2/2/9 | 1/12/7 |
| AptaGen | 0/1/5 | 2/2/6 | 36/35/27 | 0/0/6 | 2/3/6 | 10/10/13 | 2/5/9 |
| UTexas | 0/0/5 | 6/7/15 | 0/0/6 | 121/117/47 | 18/18/9 | 2/2/13 | 1/6/10 |
| RiboCentre | 2/4/12 | 1/1/7 | 2/3/6 | 18/18/9 | 85/84/37 | 0/1/7 | 1/1/11 |
| AptaDB | 0/1/7 | 2/2/9 | 10/10/13 | 2/2/13 | 0/1/7 | 257/230/88 | 20/55/14 |
| Manual | 8/2/19 | 1/12/7 | 2/5/9 | 1/6/10 | 1/1/11 | 20/55/14 | 640/515/157 |

## A  EVALUATION METRICS

For a binary classifier returning scores $s_i$ for samples with labels $y_i \in \{0, 1\}$:

**ROC-AUC.** The area under the ROC curve is

$$\text{AUC} = \int_0^1 TPR(FPR^{-1}(x)) \, dx,$$

where $TPR = \frac{TP}{TP+FN}$ and $FPR = \frac{FP}{FP+TN}$.

**Matthews correlation coefficient (MCC).** A balanced measure accounting for all four entries of the confusion matrix:

$$\text{MCC} = \frac{TP \cdot TN - FP \cdot FN}{\sqrt{(TP + FP)(TP + FN)(TN + FP)(TN + FN)}}.$$

For regression models predicting continuous affinities $\hat{y}_i$ for targets $y_i$:

**Root Mean Squared Error (RMSE).**

$$\text{RMSE} = \sqrt{\frac{1}{n}\sum_{i=1}^{n}(y_i - \hat{y}_i)^2}.$$

**Mean Absolute Error (MAE).**

$$\text{MAE} = \frac{1}{n}\sum_{i=1}^{n}|y_i - \hat{y}_i|.$$

**Coefficient of Determination ($R^2$).**

$$R^2 = 1 - \frac{\sum_{i=1}^{n}(y_i - \hat{y}_i)^2}{\sum_{i=1}^{n}(y_i - \bar{y})^2}.$$

**Pearson correlation ($r_P$).**   The linear correlation between predictions and targets:

$$r_P = \frac{\sum_{i=1}^{n}(y_i - \bar{y})(\hat{y}_i - \bar{\hat{y}})}{\sqrt{\sum_{i=1}^{n}(y_i - \bar{y})^2}\,\sqrt{\sum_{i=1}^{n}(\hat{y}_i - \bar{\hat{y}})^2}}.$$

**Spearman correlation ($r_S$).**   The rank correlation between predictions and targets:

$$r_S = 1 - \frac{6\sum_{i=1}^{n}d_i^2}{n(n^2 - 1)},$$

where $d_i$ is the difference between the ranks of $y_i$ and $\hat{y}_i$.

All metrics were computed per fold and are reported as mean $\pm$ standard deviation across cross-validation runs.

## B  ADDITIONAL BASELINE TABLES

## C  DETAILED DESCRIPTION OF THE DEEP LEARNING ARCHITECTURE

This appendix provides comprehensive technical details on the baseline deep learning model architecture for aptamer–small molecule binding prediction evaluated in this study.

### C.1  PRETRAINED ENCODERS

**Aptamer Encoder (GENA-LM)**   The aptamer sequences are encoded with the GENA-LM pretrained transformer model, which outputs token-level embeddings with hidden size 768. During training, the last two transformer layers are unfrozen and fine-tuned while earlier layers are frozen.

**Molecule Encoder (ChemBERTa)**   Molecular SMILES are encoded with the ChemBERTa pretrained transformer, also producing embeddings of hidden size 768. The last two layers are unfrozen and fine-tuned; the rest remain frozen.

### C.2  TOP LAYERS FOR EACH ENCODER

Each encoder output is further processed through a *top layer* defined as follows:

- **Identity Top Layer:** A linear projection from hidden size 768 to 255 dimensions, followed by masked mean pooling over the sequence tokens with attention mask, then layer normalization and dropout with probability 0.3.
  text

Table 6: MCC (mean ± std) on the original dataset (no negative augmentation) under grouped CV. Results are reported for three models (LGBM, MLP, RF) across aptamer encodings and molecular descriptors.

**LGBM**

| Aptamer | Morgan FP (1024) | Morgan FP (2048) | MACCS keys | RDKit descriptors | ChemBERTa |
|---|---|---|---|---|---|
| kmer(k=3) | 0.462±0.035 | 0.474±0.035 | 0.379±0.058 | 0.471±0.054 | 0.482±0.038 |
| kmer(k=4) | 0.469±0.038 | 0.455±0.039 | 0.372±0.067 | 0.479±0.026 | 0.492±0.048 |
| kmer(k=5) | 0.487±0.046 | 0.480±0.048 | 0.424±0.051 | 0.496±0.053 | 0.507±0.052 |
| onehot(L=216) | 0.478±0.039 | 0.472±0.045 | 0.438±0.054 | 0.519±0.021 | 0.536±0.037 |
| gena(mean,last) | 0.417±0.055 | 0.382±0.041 | 0.286±0.050 | 0.388±0.038 | 0.477±0.035 |

**MLP**

| Aptamer | Morgan FP (1024) | Morgan FP (2048) | MACCS keys | RDKit descriptors | ChemBERTa |
|---|---|---|---|---|---|
| kmer(k=3) | 0.401±0.070 | 0.427±0.073 | 0.173±0.092 | 0.156±0.145 | 0.388±0.036 |
| kmer(k=4) | 0.353±0.106 | 0.273±0.100 | 0.168±0.044 | 0.200±0.133 | 0.261±0.133 |
| kmer(k=5) | 0.288±0.107 | 0.288±0.122 | 0.216±0.134 | 0.186±0.050 | 0.181±0.147 |
| onehot(L=216) | 0.421±0.079 | 0.464±0.045 | 0.177±0.071 | 0.129±0.133 | 0.351±0.039 |
| gena(mean,last) | 0.154±0.069 | 0.428±0.069 | 0.240±0.082 | 0.211±0.053 | 0.295±0.120 |

**Random Forest**

| Aptamer | Morgan FP (1024) | Morgan FP (2048) | MACCS keys | RDKit descriptors | ChemBERTa |
|---|---|---|---|---|---|
| kmer(k=3) | 0.471±0.033 | 0.471±0.034 | 0.353±0.058 | 0.470±0.031 | 0.493±0.033 |
| kmer(k=4) | 0.468±0.035 | 0.468±0.037 | 0.364±0.049 | 0.466±0.037 | 0.488±0.032 |
| kmer(k=5) | 0.476±0.032 | 0.478±0.034 | 0.378±0.058 | 0.481±0.029 | 0.496±0.031 |
| onehot(L=216) | 0.471±0.034 | 0.473±0.034 | 0.381±0.056 | 0.482±0.032 | 0.499±0.030 |
| gena(mean,last) | 0.348±0.056 | 0.348±0.058 | 0.273±0.043 | 0.335±0.059 | 0.402±0.044 |

- **CNN Top Layer:** Parallel 1D convolutions with kernel sizes {3, 5, 7}, equally splitting the 255 output channels among them. Outputs concatenated are pooled globally with max pooling. Layer normalization and dropout (p=0.3) follow.

- **LSTM Top Layer:** Two-layer bidirectional LSTM with hidden state size 128 per direction (output feature size 256). Attention pooling aggregates over tokens, then a linear projection reduces to 255 dimensions, followed by normalization and dropout.

- **Transformer Top Layer:** Two Transformer encoder layers with 8 heads and feedforward dimension four times input size process the embeddings. Attention pooling aggregates token states, followed by linear projection to 255 dimensions, layer normalization, and dropout.

## C.3 FUSION LAYER

The 255-dimensional aptamer and molecule embeddings are concatenated into a 510-dimensional vector. This concatenated vector is passed through a linear layer projecting back to 255 dimensions, followed by layer normalization and dropout (p=0.3).

Although the code nominally implements gated fusion, the gating mechanism effectively reduces to simple concatenation fusion due to symmetric application of gating weights.

## C.4 CLASSIFICATION HEAD

The fusion embedding is input to an MLP classification head consisting of:

- Linear layer: 255 → 128 hidden units

- ReLU activation

- Dropout (p=0.3)

- Linear layer: 128 → 1 output logit

## C.5 Training Setup

The model is trained with the following settings:

- **Epochs:** Up to 50 epochs with early stopping based on validation Matthews correlation coefficient (MCC) with a patience of 10 epochs.
- **Batch size:** 16 samples per batch.
- **Optimizer:** AdamW with an initial learning rate of $1 \times 10^{-4}$ for all parameters.
- **Learning rate scheduling:** Cosine annealing scheduler with a reduction factor of 0.1 applied on plateau.
- **Loss function:** Binary cross-entropy with logits (BCEWithLogitsLoss).

## C.6 Effect of Fine-tuning Last Two Encoder Layers

Table 7: Effect of fine-tuning the last two transformer layers on MCC and ROC-AUC (mean ± std) across different split types.

| Split Type | Unfreeze Aptamer | Unfreeze Molecule | MCC (mean ± std) | ROC-AUC (mean ± std) |
|---|---|---|---|---|
| stratified_group | No | No | 0.376 ± 0.026 | 0.723 ± 0.018 |
| stratified_group | No | Yes | **0.411 ± 0.029** | **0.729 ± 0.018** |
| stratified_group | Yes | No | 0.316 ± 0.021 | 0.696 ± 0.021 |
| cold_aptamer | No | No | 0.373 ± 0.023 | 0.715 ± 0.026 |
| cold_aptamer | No | Yes | **0.400 ± 0.027** | **0.722 ± 0.022** |
| cold_aptamer | Yes | No | **0.289 ± 0.055** | **0.671 ± 0.037** |
| cold_molecule | No | No | 0.152 ± 0.034 | 0.555 ± 0.032 |
| cold_molecule | No | Yes | 0.151 ± 0.041 | 0.566 ± 0.034 |
| cold_molecule | Yes | No | 0.132 ± 0.039 | 0.547 ± 0.025 |