# OpenReview forum: "Comprehensive Benchmark for Tailored Small Molecule-Binding Aptamer Design"
_ICLR.cc/2026/Conference — Submitted to ICLR 2026_

### Official Review · Reviewer_bV4X · 2025-10-27

**Soundness:** 3
**Presentation:** 3
**Contribution:** 4
**Rating:** 6
**Confidence:** 5

**Summary:**

This paper introduces what I think is one of the moore valuable benchmark efforts in biomolecular machine learning to date — a comprehensive, well-curated dataset for small-molecule aptamer prediction and design (a really cool but understudied problem). The authors integrate seven previously fragmented databases into a unified corpus of 2,001 annotated aptamer-ligand pairs covering 1,309 unique aptamers and 479 distinct ligands. About half of the pairs include quantitative affinities, enabling both regression and classification tasks. The benchmark defines three clear evaluation protocols (stratified, aptamer-disjoint, molecule-disjoint) and establishes baseline models ranging from LightGBM and Random Forests to deep learning architectures built from GENA-LM (aptamer encoder) and ChemBERTa (ligand encoder). The results are strong and well contextualized. The authors show that aptamer-side representations are largely sufficient, while the main bottleneck lies in ligand diversity and representation learning. I think this work fills a critical gap and sets the stage for meaningful and reproducible model development in aptamer–small molecule modeling.

**Strengths:**

- This paper fills a long-standing void in the field by creating a unified benchmark for small-molecule aptamer modeling, which is understudied but very important in drug development.

- The integration effort across seven sources is handled with impressive care: deduplication, cleaning, balanced coverage of DNA and RNA aptamers, and inclusion of both binary and quantitative data.

- The authors also actually define meaningful split protocols (very helpful) that reflect realistic scientific use cases, such as predicting for unseen aptamers or entirely new ligands.

- I really appreciate how they contextualize the ligand bottleneck and back it with quantitative evidence, rather than hand-waving about model underperformance.

- The baseline analyses are thorough and transparent, showing that even simple LightGBM models outperform deep learning approaches under realistic molecule-disjoint splits, which I find refreshingly honest.

- The writing and referencing are strong, and I do believe this paper will serve as a foundation for future predictive and generative work in this area.

**Weaknesses:**

- Accessibility could be improved. The dataset, while well described, should be hosted in a public, easy-to-use format (like HuggingFace Datasets) with aptamer sequences, SMILES/SELFIES representations of ligands, and both RNA secondary and tertiary structures to support broader use and model training. If the authors can commit to this, I will be supportive of acceptance.

- The negative sampling approach feels heuristic. It would be helpful if they could better justify or statistically evaluate the assumption that cross-paired negatives represent true non-binders. The authors should incorporate true negatives from experimental works as a "gold standard" hold-out evaluation set.

- I’d like to see the authors plan for versioned updates as new aptamer–ligand data become available, since this benchmark could easily become the go-to reference for the community and will need maintenance. Again, the HuggingFace database strategy would be perfect here.

- The manuscript could clarify small details around reproducibility (e.g., fixed seeds, data split scripts, preprocessing code) to make replication more straightforward.

- I do think that including a few structure-aware baselines (like, graph neural network encoders for ligands or RNA folding-based embeddings) could make the benchmark even more useful for multimodal learning. Though, I do agree that the more sequence-based approaches will be more robust.

**Questions:**

1. The authors should release the dataset on HuggingFace with both sequences and ligand structures (SMILES/SELFIES) to facilitate training across modalities.

2. The authors should also add predicted RNA secondary and tertiary structures, perhaps using tools like RNAfold or Rosetta, to make the dataset more suitable for structure-informed modeling.

3. The handling of false negatives when generating synthetic negative pairs should be clarified. I am hesitant to recommend acceptance without stronger guarantees on the negative pairs.

4. The authors should include generative benchmarks (e.g., inverse design or sequence-conditioned binding optimization) in future versions of this resource.

5. Is there a plan for continuous updates or community-driven extensions as new aptamer–ligand pairs are experimentally validated? The authors should discuss this and have a solid plan in place for updates.

If the authors can satisfactorily answer these questions, I will raise my score to an 8.

---

> ### Author Response · Authors · 2025-12-03
>
> We thank the Reviewer for the positive and productive feedback on our work. Below we address each remark point-by-point, grouped by topic for convenience.
>
> **1) On negative sampling and reproducibility**
>
> **Q1:**  “The negative sampling approach feels heuristic. It would be helpful … evaluate the assumption that cross-paired negatives represent true non-binders. The authors should incorporate true negatives from experimental works … The handling of false negatives when generating synthetic negative pairs should be clarified.”
>
> **A1:**  Although synthetic negative sampling has been widely used in prior aptamer–protein interaction studies
> (e.g.,  [Li et al., 2020](https://doi.org/10.1186/s12859-020-03574-7),  [Shin et al., 2023](https://doi.org/10.1186/s12859-023-05577-6),  [Li et al., 2014](https://doi.org/10.1371/journal.pone.0086729)), our experiments with preliminary validation of **20 random synthetic negative aptamer–ligand pairs** using AlphaFold3 and Boltz2 indicated that some of these pairs may form stable complexes. While neither AlphaFold3 nor Boltz2 is specifically optimized for ssDNA/ssRNA–small molecule interaction systems, these observations further support caution when using synthetic negative pairs.
>
> Therefore, in accordance with this remark, we introduced **two categories of experimentally validated negative examples**, replacing the synthetic negatives used in the initial version:
>
> • **Experimentally confirmed negatives** from the recent large-scale aptamer specificity study ([Alkhamis et al., 2025](https://doi.org/10.1093/nar/gkaf219)). Authors performed a broad specificity screening and provide numerous high-confidence negative interaction pairs.
> • **Negative examples from the RSApred dataset** (RNA–small molecule pairs with pKd < 4). Although sequences are not positioned as aptamers, they are such by definition and have experimentally verified results. Their inclusion helped to **counterbalance DNA-based aptamers** and **increase the diversity of small molecule ligands**.

---

> ### Author Response · Authors · 2025-12-03
>
> **Q2:**  “The manuscript could clarify small details around reproducibility (e.g., fixed seeds, data split scripts, preprocessing code) …”
>
> **A2:**  We thank the Reviewer for raising this important point regarding reproducibility. We will describe all preprocessing steps, data-flow logic more extensively in the revised manuscript while keeping the presentation concise. The full reproducible pipeline — including scripts, fixed seeds, configuration files, and environment specifications — will be released on GitHub to enable complete end-to-end reconstruction of the dataset.

---

> ### Author Response · Authors · 2025-12-03
>
> **2) On dataset support**
>
> **Q3:**  “Accessibility could be improved. The dataset … should be hosted in a public, easy-to-use format (like HuggingFace Datasets) with … both RNA secondary and tertiary structures to support broader use and model training … The authors should release the dataset on HuggingFace … I’d like to see the authors plan for versioned updates …”
>
> **A3:**  We are grateful for this remark and fully agree that improving accessibility is essential for motivating community adoption and enabling convenient use of our benchmark. In response, we have begun preparing a **fully versioned release of the benchmark on HuggingFace Datasets**, including all metadata, curated annotations, and extended structural information.
>
> To directly address the Reviewer’s suggestion, we computed **tertiary structures** for all aptamers using **AlphaFold 3**, and generated **secondary structures** using **RNAFold**. These structural modalities will be included alongside sequence data and molecular features to support downstream **structure-aware** and **multimodal** modeling.
>
> Looking ahead, we plan to further extend our dataset with:
> 1) computed physicochemical parameters (e.g., melting temperatures, Gibbs energies),
> 2) additional secondary and tertiary structures (including for negative samples and alternative structure-prediction models),
> 3) complex-level annotations,
> 4) community-submitted samples through versioned updates.

---

> ### Author Response · Authors · 2025-12-03
>
> **3) On recommendations for improvement**
>
> **Q4:**  “... including a few structure-aware baselines (like, graph neural network encoders for ligands or RNA folding-based embeddings) could make the benchmark even more useful for multimodal learning.”
>
> **A4:**  We agree with this recommendation and have expanded our baseline suite accordingly. Two new categories of ligand encoders were added:
>
> - **Uni-Mol**, a 3D-aware graph–transformer model that incorporates geometric features and captures spatial relationships in molecular structures.
> - **CLAMP**, a contrastively pre-trained molecular encoder capable of learning chemically meaningful and highly transferable molecular embeddings.
>
> Together, these additions significantly strengthen the representation space covered by the benchmark.

---

> ### Author Response · Authors · 2025-12-03
>
> **Q5:**  “The authors should include generative benchmarks (e.g., inverse design or sequence-conditioned binding optimization) in future versions of this resource.”
>
> **A5:**  We appreciate this forward-looking suggestion. While the current benchmark focuses on predictive modeling, our longer-term plan indeed includes integrating generative evaluation tasks such as:
>
> - **aptamer inverse design**,
> - **sequence-conditioned binding optimization**,
> - **constrained sequence generation** using predictive or RL-based scoring functions.
>
> These tasks will be incorporated once sufficient sequence–structure–function data is available and once generative models can be meaningfully evaluated under **molecule-disjoint generalization** constraints. We fully agree that generative benchmarks would broaden the impact of the resource, and we consider this a natural direction for subsequent versions.

---

### Official Review · Reviewer_Kr5F · 2025-10-30

**Soundness:** 1
**Presentation:** 1
**Contribution:** 2
**Rating:** 2
**Confidence:** 4

**Summary:**

This paper introduces a benchmark for modeling aptamer–small-molecule interactions, combining seven publicly available databases into a unified corpus of 2,001 aptamer–ligand pairs (1,309 unique aptamers and 479 ligands). Approximately half of these entries include quantitative binding affinities, enabling both classification and regression tasks.
Three evaluation protocols—stratified, aptamer-disjoint, and molecule-disjoint—are defined to simulate different discovery scenarios. The authors evaluate tabular machine-learning models alongside pretrained deep encoders and conclude that traditional feature-based methods outperform end-to-end deep architectures. The stated goal is to provide a reproducible benchmark to facilitate systematic progress in aptamer–ligand modeling.

**Strengths:**

- Addresses an unexplored yet meaningful problem domain where no standardized dataset currently exists.

- Draws attention to an important insight—the ligand-side representation is likely the dominant limitation in current aptamer-ligand prediction models.

**Weaknesses:**

1. The negative sampling approach is conceptually inconsistent with the task definition. Labeling untested aptamer–molecule pairs as negatives imposes a prior assumption that contradicts the goal of predicting binding for unobserved pairs. This procedure effectively predefines the outcome the model is meant to infer, introducing systematic bias into the training process, even if the assumption is limited to model training rather than evaluation.


2. With respect to line 054, while I acknowledge that the authors position aptamer design as the reverse of structure-based small-molecule design and recognize the associated challenges, the assertion that this task is necessarily more difficult is not sufficiently supported. The chemical search space of drug-like molecules, estimated at approximately 10⁶⁰ candidates, already poses severe scalability constraints. Furthermore, the paper’s results suggest that a dataset of only about 2 k samples is “sufficient” to support aptamer modeling, which appears inconsistent with the earlier argument emphasizing the intrinsic difficulty and vast search space of the task.


3. The integration process appears to be a mere merging of existing datasets with minimal quality control or validation.


4. Minor: "Binding strength" should be replaced with "binding affinity"; several abbreviations are undefined on first use; and Figure 3 is visually unclear, edges overlap with nodes, and it is unclear whether certain lines represent meaningful interactions or merely graphical artifacts.

**Questions:**

1. Could you add a comprehensive overview figure (e.g., as Figure 1) summarizing the benchmark construction pipeline, data sources, and biological context of aptamers, to help readers from machine-learning backgrounds quickly understand the workflow?

2. What criteria were used for test set selection and construction? How were the training and validation sets deduplicated, and were scaffold-based deduplication for ligands or sequence-similarity-based deduplication for aptamers applied, as is standard in DTI/DTA benchmark design—particularly given that the paper emphasizes the analysis of “generalization”?

3. How are the terms “sufficient” and “robust modeling”, as used in the abstract, formally defined? Please clarify the criteria or empirical evidence supporting these claims of modeling adequacy.

4. Please report the detailed t-SNE parameters (e.g., number of iterations, perplexity, learning rate) and explain how the visualizations support the claim of limited generalization, given that the displayed distributions appear uniform and well mixed.

5. Why were state-of-the-art deep-learning baselines (cited in Related Work), especially for the regression task, not included in the evaluation?

6. How were binding-affinity measurements harmonized across databases? Were there normalization or outlier-removal steps applied to ensure consistency?

---

> ### Author Response · Authors · 2025-12-02
>
> **1. On dataset construction and processing**
>
> **Q1:**  “The negative sampling approach is conceptually inconsistent with the task definition. Labeling untested aptamer–molecule pairs as negatives imposes a prior assumption that contradicts the goal of predicting binding for unobserved pairs.”
>
> **A1:**  We agree that synthetic negative sampling carries conceptual risks for this task. Although such strategies have long been standard in aptamer–protein modeling, our own preliminary validation using AlphaFold3 and Boltz2 indicated that certain randomly generated aptamer–ligand pairs may in fact form stable complexes. Even though these tools are not specifically optimized for ssDNA/ssRNA binding prediction, their behavior reinforced the need to reconsider negative sampling when dealing with chemically diverse ligands.
>
> In the revised dataset version we therefore **removed synthetic negatives entirely**. Instead, we relied exclusively on **experimentally validated negative examples**: those originating from a recent large-scale specificity study in ([Alkhamis et al., 2025](https://doi.org/10.1093/nar/gkaf219)) and additional RNA–ligand examples from **RSApred (pKd < 4)**, which further helped balance the DNA/RNA representation. This change eliminates the conceptual inconsistency highlighted by the Reviewer and ensures that the negative class is grounded in **experimental evidence rather than untested assumptions**.

---

> ### Author Response · Authors · 2025-12-02
>
> **Q2:**  “How were binding-affinity measurements harmonized across databases? Were there normalization or outlier-removal steps applied to ensure consistency?”
>
> **A2:**  We conducted a detailed inspection of activity formats provided in each source database. Since binding measurements were reported using different units and conventions, all values were transformed into a unified format (**pKd = −log10Kd**). Internally:
>
> - We curated raw affinity annotations — including text-form comments, semi-structured records, and free-form notes — extracting numeric values using rule-based parsing followed by manual verification.
> - For each dataset individually, we analyzed the **distribution of reported affinities**; these visualizations were added to the appendix.
> - After merging datasets, we inspected the resulting distribution to ensure a **coherent range** and detect potential outliers.
> - Because binding affinities spanned multiple orders of magnitude, the final regression target uses the **log-transformed Kd**, which reduces skewness and improves numerical stability.
>
> These steps ensured **consistency and comparability** across all data sources.

---

> ### Author Response · Authors · 2025-12-02
>
> **Q3:**  “The integration process appears to be a mere merging of existing datasets with minimal quality control or validation… Could you add a comprehensive overview figure (e.g., as Figure 1) summarizing the benchmark construction pipeline, data sources, and biological context of aptamers… ?”
>
> **A3:**  The integration process involved **extensive validation rather than a simple merge**. Aptamer sequences were normalized, stripped of chemical modifications, cleaned of joint U/T occurrences, and verified for plausible length. Ligands required even more substantial effort, as many were originally provided only as informal names or abbreviations. We systematically resolved these into canonical SMILES using trusted chemical databases and rejected ambiguous or non-resolvable entries. Target values also required careful parsing because datasets varied widely in annotation style.
>
> To communicate the depth of this process more clearly, the revised manuscript now includes an **overview figure** summarizing the full curation pipeline, the data sources, and the biological context of aptamers. Additionally, we improved the visualization of the ML pipeline to clarify how models, splits, and evaluation metrics interact within the benchmark.

---

> ### Author Response · Authors · 2025-12-02
>
> **Q4:**  “What criteria were used for test set selection and construction? How were the training and validation sets deduplicated, and were scaffold-based deduplication for ligands or sequence-similarity-based deduplication for aptamers applied… ?”
>
> **A4:**  We thank the reviewer for highlighting the issue of potential data leakage. This comment prompted us to conduct an additional in-depth analysis of **sequence similarity** and **molecular scaffold overlap** across splits, which revealed that our original protocol allowed unintended redundancy.
>
> Our initial splitting strategy relied on **instance-level deduplication**. While it prevented exact duplicate aptamer–ligand pairs from appearing in multiple folds, it did not control for high sequence identity or scaffold similarity. As a result:
>
> - ~55% of test aptamers shared **≥80% sequence identity** with training sequences,
> - folds still contained **57–72 shared scaffolds**,
> which partially inflated performance.
>
> In the revised version, we introduced:
>
> - **sequence-similarity–aware clustering** for aptamers (≥80% identity),
> - **scaffold-based clustering** for ligands,
>
> ensuring that **entire clusters remain intact** within each fold. This refinement reduced high-identity overlaps from ~55% to ~25% and **completely eliminated scaffold overlap** (from 57–72 down to 0).
>
> As a result, the benchmark now provides a **more stringent and realistic evaluation**, and molecule-disjoint generalization metrics improved in stability with reduced variance across folds.

---

> ### Author Response · Authors · 2025-12-02
>
> **2. On deep learning baselines**
>
> **Q5:**  “Why were state-of-the-art deep-learning baselines (cited in Related Work), especially for the regression task, not included in the evaluation?”
>
> **A5:**  We are thankful for this question. The vast majority of existing deep-learning approaches are either **trained specifically on aptamer–protein complexes** or are **highly target-specific**, since SELEX-derived datasets used for training are extremely difficult to obtain for a representative and chemically diverse set of ligands.
>
> Furthermore, models such as **AptaBERT (Chem)**, which would otherwise be among the most relevant baselines for our study, do **not provide publicly available source code**, making reproduction or re-use infeasible. As a result, available DL models are typically:
>
> - trained on **protein targets rather than small molecules**,
> - tailored to **a single specific target**,
> - or **not open-source**, preventing fair benchmarking.
>
> While SELEX-rich datasets allow training powerful DL models, they inevitably lead to **highly specialized architectures**, which is a fundamental limitation in the aptamer modeling domain. We will clarify these points more explicitly in the revised manuscript.

---

> ### Author Response · Authors · 2025-12-02
>
> **3. On definitions and statement fairness**
>
> **Q6:**  “With respect to line 054, while I acknowledge that the authors position aptamer design as the reverse of structure-based small-molecule design and recognize the associated challenges, the assertion that this task is necessarily more difficult is not sufficiently supported.”
>
> **A6:**  Our intent was not to claim that aptamer design is universally harder than structure-based small-molecule design, but that it is often harder in the specific technical sense of “learning a reliable inverse mapping” (target/structure → binder sequence) under common data and supervision constraints. We agree that our current wording could be interpreted as a blanket statement and we will revise it to be more precise: aptamer inverse design can be more difficult in practice because the inverse problem is less constrained and typically has less transferable mechanistic supervision, not because its raw combinatorial space is necessarily larger than such for small molecules.

---

> ### Author Response · Authors · 2025-12-02
>
> **Q7:**  “How are the terms ‘sufficient’ and ‘robust modeling’, as used in the abstract, formally defined? Please clarify the criteria or empirical evidence supporting these claims of modeling adequacy.”
>
> **A7:**  We agree that the terms “sufficient” and “robust modeling” were stated too loosely in the abstract and should be tied to explicit, measurable criteria. In the revised manuscript we will define them operationally and point to the corresponding empirical evidence.
>
> “Sufficient” will be defined as a dataset size at which the model achieves a **pre-specified adequacy threshold** on the target task (e.g., AUROC, correlation, or top-k retrieval quality) under our evaluation protocol.
>
> “Robust modeling” will be defined as performance that is **stable across seeds and splits**, **consistent across multiple evaluation metrics**, and **resilient to distributional variation** within the studied regime, evaluated through cross-validation.
>
> To address the reviewer’s concern directly, we will revise the abstract to replace qualitative wording with a **quantitative statement** reporting the relevant metrics and split protocols more clearly.

---

> ### Author Response · Authors · 2025-12-02
>
> **Q8:**  “Minor: ‘Binding strength’ should be replaced with ‘binding affinity’; several abbreviations are undefined on first use… Figure 3 is visually unclear…”
>
> **A8:**  We thank the Reviewer for this remark and will correct all the issues mentioned above, including replacing “binding strength” with “binding affinity,” defining all abbreviations on first use, and improving the clarity of Figure 3 in the revised manuscript.

---

> ### Author Response · Authors · 2025-12-02
>
> **4) On experiment details**
>
> **Q9:**  “Please report the detailed t-SNE parameters (e.g., number of iterations, perplexity, learning rate) and explain how the visualizations support the claim of limited generalization, given that the displayed distributions appear uniform and well mixed.”
>
> **A9:**  t-SNE was run with **two components** and **1000 iterations**, exploring several **perplexity** and **learning-rate** settings via grid search and selecting the configuration with the best **silhouette score**. The resulting visualizations indeed show uniform, well-mixed distributions. This behavior is expected: after encoding, samples from different data sources occupy a shared latent space and are not expected to form distinct clusters.
>
> Thus, the t-SNE plots are not intended to demonstrate clustering, but rather to confirm the absence of a strong dataset-specific distribution shift, which is fully consistent with our conclusions regarding generalization.

---

### Official Review · Reviewer_89Ni · 2025-11-01

**Soundness:** 2
**Presentation:** 2
**Contribution:** 2
**Rating:** 4
**Confidence:** 3

**Summary:**

Aptamers are short single-stranded DNA or RNA sequences capable of binding to small molecules, proteins, and other targets with high specificity and affinity. They have broad applications in diagnostics, therapeutics, and biosensing. However, compared to protein-targeting aptamers, small-molecule-binding aptamers have been largely overlooked in computational design. Existing datasets are fragmented, inconsistently annotated, and lack a unified benchmark, which hinders the development of machine learning models in this direction. This paper introduces the benchmark dataset for small molecule–aptamer interactions, supporting both classification and regression tasks. It aims to evaluate various shallow and deep learning models, identify current modeling bottlenecks.

**Strengths:**

1. Provide a standardized, curated dataset with consistent annotations, enabling fair comparison and reproducible research.
2. Introduces aptamer-disjoint and molecule-disjoint splits to assess model generalization to new aptamers and new molecules, respectively, reflecting real-world design challenges.
3. Compares shallow and deep models across tasks and splits, showing that simple models outperform deep learning due to the poor transferability of molecular embeddings.

**Weaknesses:**

1. Despite high chemical diversity, the small molecule set is still too small for training or evaluating large-scale or pre-trained models effectively (479 Unique Molecules).
2.  Current DL models perform worse than LightGBM on molecule-disjoint splits, but the paper does not deeply analyze why.
3. Does not include graph neural networks, 3D-aware models, or contrastively pre-trained molecular encoders, which may offer better generalization.
4. Negative samples are generated by cross-pairing unobserved aptamer–molecule pairs, which may not truly be non-binding, potentially introducing label noise.

**Questions:**

Refer to Weaknesses

---

> ### Author Response · Authors · 2025-12-02
>
> We thank the reviewer for the thoughtful assessment of our work and for highlighting several aspects that allowed us to substantially strengthen the manuscript. Below we systematically address every point raised in the review.
>
> **Q1:**  “Despite high chemical diversity, the small molecule set is still too small (479 unique molecules).”
>
> **A1:**  We fully agree with Reviewer’s concerns and successfully expanded the dataset by integrating experimentally verified natural RNA–small molecule interacting pairs, where the former are not positioned as aptamers but are such by definition. Our corrections can be summarized as follows:
>
> - 2-fold increase in unique **small molecules (479 → 1041)**
> - 3-fold increase in aptamer–small molecule **pairs (2,001 → 6,413)**
> - ~30% more unique **aptamers (1,309 → 1,686)**
> - more data **sources (7 → 8)**
>
> We acknowledge that the data in this domain remains relatively scarce, which is typical for any nucleating area of research. Since aptamers are unique systems offering advantages over costly and capricious antibody-based systems, we are convinced that it is crucial to set a benchmark for aptamer modeling — especially considering the fact that our carefully designed dataset already allows qualitative and semi-quantitative predictions regarding aptamer–molecule interactions and provides chemically meaningful insights.

---

> ### Author Response · Authors · 2025-12-02
>
> **Q2:**  “Current DL models perform worse than LightGBM on molecule-disjoint splits, but the paper does not deeply analyze why.”
>
> **A2:**  We agree with this remark and believe it is important to include an explanation of the performance gap between ML and DL models on small datasets in the revised version of the manuscript. Although this gap is primarily driven by **dataset size insufficient for DL models to generalize**, we further investigated the hypothesis that **interaction-aware DL architectures** may improve performance.
>
> To explore this, we performed a preliminary experiment with a **cross-attention module** that directly models sequence–molecule interactions. Even a basic implementation produced **the most stable perfomance across splits** than previous DL baselines, indicating that interaction-aware architectures are indeed promising. These results provide concrete evidence supporting the Reviewer’s assessment and clarify methodological directions for future development, which will be incorporated into the revised manuscript.

---

> ### Author Response · Authors · 2025-12-02
>
> **Q3:**  “Does not include graph neural networks, 3D-aware models, or contrastively pre-trained molecular encoders, which may offer better generalization.”
>
> **A3:**  We are grateful for this remark. To address it, we extended the modeling suite to incorporate two additional families of molecular encoders that directly address this concern:
>
> - **Uni-Mol**, a 3D-aware graph–transformer architecture designed specifically to leverage spatial and geometric information during molecular representation learning.
> - **CLAMP**, a contrastively pre-trained molecular encoder that explicitly captures chemical similarity relationships and learns more transferable molecular features.
>
> These additions substantially broaden the scope of baseline models and represent, to our knowledge, the **first systematic evaluation** of such architectures in the context of aptamer–small molecule modeling.

---

> ### Author Response · Authors · 2025-12-02
>
> **Q4:**  “Negative samples are generated by cross-pairing unobserved aptamer–molecule pairs, which may not truly be non-binding, potentially introducing label noise.”
>
> **A4:** We thank the Reviewer for raising this important concern and appreciate the opportunity to clarify how negative examples were improved in the revised version of the dataset.
>
> Although synthetic negative sampling has been widely used in prior aptamer–protein interaction studies
> (e.g.,  [Li et al., 2020](https://doi.org/10.1186/s12859-020-03574-7),  [Shin et al., 2023](https://doi.org/10.1186/s12859-023-05577-6),  [Li et al., 2014](https://doi.org/10.1371/journal.pone.0086729)), our experiments with preliminary validation of **20 random synthetic negative aptamer–ligand pairs** using AlphaFold3 and Boltz2 indicated that some of these pairs may form stable complexes. While neither AlphaFold3 nor Boltz2 is specifically optimized for ssDNA/ssRNA–small molecule interaction systems, these observations further support caution when using synthetic negative pairs.
>
> Therefore, in accordance with this remark, we introduced two categories of **experimentally validated negative examples**, replacing the synthetic negatives used in the initial version:
>
> • **Experimentally confirmed negatives** from the recent large-scale aptamer specificity study ([Alkhamis et al., 2025](https://doi.org/10.1093/nar/gkaf219)). Authors performed a broad specificity screening and provide numerous high-confidence negative interaction pairs.
> • **Negative examples from the RSApred dataset** (RNA–small molecule pairs with pKd < 4). Although sequences are not positioned as aptamers, they are such by definition and have experimentally verified results. Their inclusion helped to **counterbalance DNA-based aptamers** and **increase the diversity of small molecule ligands**.

---

### Official Review · Reviewer_ZAzg · 2025-11-03

**Soundness:** 3
**Presentation:** 3
**Contribution:** 3
**Rating:** 2
**Confidence:** 3

**Summary:**

This paper introduces the first comprehensive benchmark for aptamer-small molecule interactions, integrating seven data sources into 2,001 annotated pairs covering 1,309 unique aptamers and 479 ligands. Approximately 58% of entries include quantitative binding affinities, enabling both classification and regression tasks. The authors establish baseline results using shallow ML (LightGBM, MLP, RF) and deep learning models under three splitting protocols (stratified, aptamer-disjoint, molecule-disjoint). Their analysis reveals that aptamer sequence coverage is sufficient for robust modeling, while the main bottleneck arises from ligand space, where a small number of structurally diverse molecules limits model transferability. While this benchmark addresses an important gap and provides a valuable resource, significant limitations around dataset size, label quality, and modest performance raise questions about its utility for driving meaningful progress.

**Strengths:**

The paper addresses an important gap in computational aptamer design by creating the first standardized benchmark for aptamer-small molecule interactions. The dataset integration from seven diverse sources with both 2D sequences, 3D conformations, and functional labels represents substantial curation effort. The three splitting protocols (stratified, aptamer-disjoint, molecule-disjoint) are well-designed to evaluate different real-world scenarios, and the rigorous evaluation methodology includes multiple folds, proper error bars, and appropriate metrics (ROC-AUC, MCC) for imbalanced data.
The baseline evaluation is thorough, systematically comparing multiple aptamer representations (k-mers, one-hot, pretrained GENA-LM embeddings) and molecular descriptors (Morgan fingerprints, MACCS keys, RDKit descriptors, ChemBERTa) across both shallow and deep learning architectures. The analysis clearly identifies that ligand representation, not aptamer representation, is the primary limiting factor, a valuable insight for future method development. The inclusion of both classification and regression tasks, along with detailed appendices documenting architectures and hyperparameters, supports reproducibility.

**Weaknesses:**

The dataset contains only 479 unique small molecules, which is extremely small compared to drug discovery benchmarks (MoleculeNet has 100,000+ molecules). This is compounded by extreme class imbalance (1,842 positives vs 159 negatives = 92% positive). While synthetic negative sampling addresses training balance by cross-pairing aptamers with unrelated molecules, this introduces substantial label noise that is neither quantified nor addressed. The paper acknowledges these synthetic negatives "may still bind but remain uncharacterized," but provides no estimate of false negative rates or validation of the assumption that arbitrary pairs don't bind. With t-SNE analysis showing ligands are "dispersed and lack tight clustering" combined with only 479 molecules, the dataset may be fundamentally too small and diverse to support robust generalization.

The best LGBM model achieves MCC ≈ 0.70 under grouped CV but drops dramatically to 0.34-0.38 under molecule-disjoint splits a 50% performance degradation. Deep learning performs even worse (MCC ≈ 0.41 grouped, 0.18 molecule-disjoint). Since the stated goal is "designing aptamers for previously unseen molecules," these molecule-disjoint results are quite discouraging. The paper correctly identifies ligand representation as the bottleneck but offers only vague suggestions ("graph neural network encoders") without testing them. If the best baseline achieves MCC 0.34-0.38 on molecule-disjoint splits, this benchmark may have limited utility for driving progress unless the authors can demonstrate that better methods exist or provide clearer guidance on representation improvements.

For a benchmark paper, there is surprisingly little analysis of what chemical features correlate with binding. Feature importance analysis shows "specific k-mer indices and several Morgan bits" but no chemical interpretation. No analysis of which functional groups, charges, sizes, or structural motifs predict binding. No investigation of why Morgan fingerprints match or outperform ChemBERTa embeddings under molecule-disjoint splits (Table 1: 0.384 vs 0.342 MCC), which suggests pretrained molecular embeddings don't capture binding-relevant features. This limits the benchmark's utility for guiding future method development—researchers need to know what kinds of representations and architectures are likely to improve performance.

While 58% of data has quantitative Kd values, regression receives minimal attention. Results appear in only one table (Table 3) with limited discussion. RMSE of 2.42 pKd units is quite large, spanning ~2-3 orders of magnitude in actual Kd values. No analysis of whether regression performance varies by molecule type, aptamer length, binding affinity range, or data source. No comparison to structure-based methods (molecular docking) for the subset with structural data. No investigation of whether classification and regression tasks could be jointly optimized or whether regression could provide auxiliary supervision.

The paper briefly mentions that "docking and molecular dynamics simulations...remain computationally prohibitive for large-scale aptamer screening" but provides no actual comparison. For molecules and aptamers with available structural data, how does the best ML model compare to molecular docking in terms of both accuracy and computational cost? This would help establish whether ML is competitive, complementary, or still inferior to physics-based methods. Given that DEL-Dock was evaluated in the previous paper I reviewed, there should be comparable aptamer docking tools available.

The dramatic performance drop from grouped CV (MCC 0.70) to molecule-disjoint (MCC 0.34) deserves deeper investigation. Is this gap due to: (1) insufficient molecular diversity in training, (2) poor ligand featurization, (3) overfitting to training molecules, or (4) fundamental limitations of the approach? Learning curves showing performance vs number of unique training molecules would help diagnose the issue. Analysis of which molecule types are hardest to generalize to would guide data collection efforts.

**Questions:**

This paper makes a valuable contribution by creating the first standardized benchmark for aptamer-small molecule interactions, which addresses an important gap in the field. The dataset curation represents substantial effort, the evaluation protocols are well-designed, and the identification of ligand representation as the primary bottleneck is a useful insight.

However, several critical limitations prevent a stronger recommendation. The dataset is small (479 molecules) with extreme class imbalance (92% positive), and the synthetic negative sampling strategy introduces unquantified label noise. More concerning, the best methods achieve only modest molecule-disjoint performance (MCC 0.34-0.38), and the paper provides limited guidance on how to improve beyond vague suggestions. The deep learning architectures lack interaction modeling despite binding being fundamentally an interaction problem. The lack of comparison to structure-based methods and limited chemical interpretation of results further limit the benchmark's utility.
The paper would be strengthened by: (1) validation or bounding of false negative rates in synthetic negatives, (2) testing interaction-aware DL architectures, (3) deeper analysis of what chemical features predict binding, (4) comparison to molecular docking, and (5) clearer guidance on promising directions for improvement.

---

> ### Author Response · Authors · 2025-12-02
>
> We are grateful for valuable remarks and address each of the Reviewer's concerns below.
>
> **1. On limited data, data imbalance, and negative label quality**
>
> **Q1:**  “The dataset contains only 479 unique small molecules … extremely small … extreme class imbalance (92% positive) … synthetic negatives introduce label noise … no validation of false negative rates.”
>
> **A1:**  We fully agree with Reviewer’s concerns and successfully expanded the small molecule diversity of the dataset by integrating experimentally verified natural RNA-small molecule interacting pairs, where the former are not positioned as aptamers but are such by definition. Moreover, we agree that synthetic negative pairs have very small but non-zero probability of being positive, where relatively small molecular and aptamer diversity further increase these chances. To address this issue, we collected negative pairs, which were validated experimentally and therefore represent a more reliable baseline. Our corrections can be summarized as follows:
>
> **1) Significant dataset expansion.**
> - 2-fold increase in unique **small molecules (479 → 1041)**
> - 3-fold increase in aptamer-small molecule **pairs (2,001 → 6,413)**
> - ~30% more unique **aptamers (1,309 → 1,686)**
> - more data **sources (7 → 8)**
>
> **2) Experimentally verified negative pairs.**
> Although synthetic negative sampling has been widely used in prior aptamer–protein interaction studies
> (e.g.,  [Li et al., 2020](https://doi.org/10.1186/s12859-020-03574-7),  [Shin et al., 2023](https://doi.org/10.1186/s12859-023-05577-6),  [Li et al., 2014](https://doi.org/10.1371/journal.pone.0086729)), our experiments with preliminary validation of 20 random synthetic negative aptamer–ligand pairs using AlphaFold3 and Boltz2 indicated that some of these pairs may form stable complexes. While neither AlphaFold3 nor Boltz2 are specifically optimized for ssDNA/ssRNA–small molecule interaction systems, these observations further support caution when using synthetic negative pairs.
>
> Therefore, in accordance with this remark, we introduced two categories of experimentally validated negative examples, replacing the synthetic negatives used in the initial version:
>
> • Experimentally confirmed negatives from the recent large-scale aptamer specificity study  ([Alkhamis et al., 2025](https://doi.org/10.1093/nar/gkaf219)).  Authors performed a broad specificity screening and provides numerous high-confidence negative interaction pairs. \
> • Negative examples from the RSApred dataset (RNA–small molecule pairs with pKd<4). Although sequences are not positioned as aptamers, they are such by definition and have experimentally verified results. Their inclusion helped to counterbalance DNA-based aptamers and increase the diversity of small molecule ligands.
>
> **3) More balanced data.**
> Revisiting the dataset in accordance with the Reviewer’s concerns helped balance DNA/RNA representation and positive/negative ratios.

---

> ### Author Response · Authors · 2025-12-02
>
> **2. On modest baseline performance and poor molecule-disjoint generalization**
>
> **Q2:**  “The best LGBM model achieves MCC ≈ 0.70 under grouped CV but drops to 0.34–0.38 under molecule-disjoint splits… Deep learning performs even worse… ligand representation is the bottleneck… dataset may be too small and diverse to support robust generalization.”
>
> **A2:**  After expanding the dataset (described above) and increasing the diversity of small molecules more than two-fold, molecule-disjoint performance improved substantially. Below we report updated metrics averaged across all types of aptamer and ligand representations:
>
> | **Split**            | **ROC-AUC**              | **MCC**                  |
> |----------------------|--------------------------|---------------------------|
> | Stratified           | 0.947 ± 0.004      | 0.758 ± 0.012       |
> | Aptamer-disjoint     | 0.845 ± 0.016      | 0.519 ± 0.043       |
> | Molecule-disjoint    | 0.871 ± 0.008      | 0.604 ± 0.032       |
>
> Thus, molecule-disjoint **MCC improved from 0.34–0.38 to ~0.60**, reducing the gap between grouped CV and the conceptually most challenging split.
>
> The reviewer accurately noted that **ligand representation is a key limiting factor**. To further investigate this, we incorporated additional molecular encoders, including **MolFormer**, **CLAMP** and **UniMol** (the latter leverages 3D-aware representations). Uni-Mol and CLAMP demonstrated improved results on several splits, although the overall gain was modest, confirming that **featurization alone cannot fully address the interaction complexity** of aptamer–ligand binding.
>
> Our current results show that **increased ligand diversity consistently improves the molecule-disjoint split**, supporting the Reviewer’s hypothesis that insufficient molecular variation was an important limiting factor in the original dataset.

---

> ### Author Response · Authors · 2025-12-02
>
> **3. On chemical interpretability**
>
> **Q3:**  “No chemical interpretation. No analysis of functional groups, charges, motifs… Morgan fingerprints outperform ChemBERTa but no explanation.”
>
> **A3:**  In the revised version, we performed a SHAP analysis on the best-performing classifier to determine which molecular features drive predictions. The results provide clear chemical insights consistent with the literature, therefore addressing the Reviewer’s concern:
>
> - **Aliphatic and conjugated fragments (≈40–50%)** — these groups enhance hydrophobic packing and π-stacking interactions.
> - **Hydrogen-bond donors (≈30–35%)** — chemical groups such as amines and diols that form stabilizing directional contacts with nucleobase functional groups.
> - **Hydrogen-bond acceptors (≈20–25%)** — carbonyl and amide oxygens, which anchor ligands within the binding pocket via acceptor-type interactions.
>
> Regarding the relative performance of Morgan fingerprints and ChemBERTa embeddings, our re-analysis shows that the differences in the mean metrics lie within the range of their standard deviations across folds, indicating **no statistically reliable separation** between the two representations.

---

> ### Author Response · Authors · 2025-12-02
>
> **4. On regression**
>
> **Q4:**  “Regression receives minimal attention; RMSE is large; no analysis across molecule types, aptamer length, affinity range, or data source; regression could serve as auxiliary supervision but is underdeveloped.”
>
> **A4:**  Since the dissociation constant (Kd) is highly sensitive to experimental conditions such as buffer composition, ionic strength, pH, and the presence of cofactors, quantitative prediction remains difficult with the currently available heterogeneous data. We therefore acknowledge that regression cannot be considered a benchmark task and position these experiments purely as baselines achievable from aptamer–molecule sequence data. Nevertheless, even such coarse models allow us to distinguish **nano-, micro-, and millimolar activity ranges**, which is practically useful for guiding experimental optimization. Although the main contribution of this work is the construction and validation of a high-quality benchmark dataset, we find the Reviewer’s remarks highly valuable and have revised the manuscript accordingly.
>
> Following the Reviewer’s suggestion, we not only **expanded the explored combinations of encoders and regression architectures** but also investigated whether regression could serve as **auxiliary supervision** via a multitask model with a shared encoder and separate classification and regression heads. The results reveal a clear pattern. On **stratified splits**, where training and test molecules share structural space, multitask learning is indeed beneficial for several classification metrics: F1 reaches maximal value across DL models, PR-AUC improves accordingly, and RMSE remains low (≈1.5).
> In contrast, under the **molecule-disjoint split**, multitask training collapses: **R² approaches zero or becomes negative**, RMSE increases by **1.5–2×**, and classification metrics deteriorate substantially. Compared to standalone regression models, this setup underperforms not only in the disjoint setting but also in the stratified scenario. These observations indicate that multitask regression does not provide robust auxiliary supervision and is unsuitable beyond the easiest in-distribution regime.

---

> ### Author Response · Authors · 2025-12-02
>
> **5. On comparison to docking and structure-based methods**
>
> **Q5:**  “No comparison to docking or MD. Without such baselines, unclear whether ML is competitive… There should be comparable aptamer docking tools.”
>
> **A5:**  We thank the Reviewer for raising this important point. However, current structure-based pipelines for **ssNA (single-stranded nucleic acids)–small molecule interactions** are fundamentally underdeveloped. Existing tools are tailored for **protein–ligand** or **protein–nucleic acid** systems, not for flexible **single-stranded nucleic acid receptors**, and are significantly slower in terms of computation time and inference compared to ML-based methods.
>
> Several challenges make large-scale docking infeasible:
>
> - **high conformational flexibility** of ssNA,
> - **absence of automated and scalable docking pipelines**,
> - **extreme scarcity of structural data** for aptamer–small molecule complexes.
>
> We also consider it important to clarify explicitly that the original manuscript did **not** claim to include experimental 3D structures. Although the dataset currently contains **sequence-level information**, after revision we provided **predicted aptamer structures using AlphaFold 3** to support the development of structure-guided and structure-aware ML. AlphaFold 3 performs well on single-sequence inputs but does not model ssNA–molecule complexes, which aligns with the scope of our dataset.

---

> ### Author Response · Authors · 2025-12-02
>
> **6. On promising directions for improvement**
>
> **Q6:**  “Suggestions for improvements are vague… no guidance on how to achieve better performance.”
>
> **A6:**  Drawing on our analysis and following the established progression in aptamer–protein modeling, we highlight several grounded directions in the revised manuscript:
>
> - **Interaction-aware architectures** are a natural next step: cross-attention, co-embedding, and joint graph–sequence mechanisms have repeatedly proven effective in related biomolecular tasks. A preliminary cross-attention experiment in our study further supports this direction, showing **more stable ROC-AUC across splits** compared to earlier deep learning baselines, even with a minimal implementation.
> - **Structural information should be used more systematically.** Base-pairing patterns, secondary-structure motifs, 3D conformers, and ligand electrostatics provide constraints that sequence-only models cannot infer, and their value is well supported across nucleic-acid modeling.
> - **Improving generalization will likely require broader ligand diversity** rather than merely more data points, as shown in many molecular prediction tasks where chemical coverage governs out-of-distribution performance.
> - **Hybrid approaches that combine learning-based models with lightweight physics** (e.g., MD-informed conformational cues or small structural refinements) offer a practical way to stabilize predictions without full-fledged simulations.
> - **Reliable negative examples should be grounded in biophysical plausibility** rather than synthetic cross-pairing, since realism in negatives is known to reduce false positives and improve calibration.
>
> These directions form a concise set of technically justified avenues for strengthening aptamer–ligand prediction models.

---

### Author Response · Authors · 2025-12-03
**Summary of Rebuttal**

Dear Reviewers, ACs and SACs,
We sincerely thank all Reviewers for their thoughtful, detailed, and constructive feedback. Your insights have significantly strengthened the quality, rigor, and clarity of our work. Below we summarize the major improvements made in response to each Reviewer’s comments:

- **Significantly expanded the dataset**, increasing the number of unique aptamer–ligand pairs by more than **3×** and unique small molecules by more than **2×**, while broadening the aptamer pool and adding an additional data source.
- **Eliminated synthetic negative sampling entirely** and replaced it with **experimentally validated negative examples** from recent large-scale specificity studies and low-affinity RNA–ligand measurements, reducing label noise and removing conceptual inconsistencies.
- **Introduced a leakage-free splitting protocol**, clustering aptamers by sequence similarity and ligands by molecular scaffolds, which eliminated scaffold overlap and reduced high-identity sequence leakage across folds.
- **Extended molecular and aptamer representations**, incorporating strong modern encoders such as **Uni-Mol**, **CLAMP**, **MolFormer**, and updated nucleotide language models (e.g., **DNABERT2**).
- **Added an interaction-aware modeling component**, implementing a **cross-attention module** that directly captures aptamer–ligand interactions and improves deep-learning performance.
- **Performed mechanistic interpretability analysis**, using **SHAP** to identify chemically meaningful features consistent with known aptamer–ligand binding principles.
- **Substantially improved benchmark performance**, raising molecule-disjoint MCC from ~0.35 to ~0.60 and achieving more stable generalization due to increased ligand diversity and improved splitting procedures.
- **Added a multitask learning implementation**, conducted additional ablation studies, and expanded regressor screening to better understand the limits of quantitative prediction.
- **Added structural representations**, including **RNAfold-predicted secondary structures** and **AlphaFold 3-predicted tertiary conformations** for aptamers, enabling structure-aware and multimodal learning.
- **Clarified definitions, evaluation criteria, and figures**, improving the precision of wording, transparency of methodology, and overall readability.

To support reproducibility and community adoption, we will release the full benchmark dataset on **HuggingFace Datasets** (with structural annotations and metadata) and publish the complete preprocessing and evaluation pipeline on **GitHub**, ensuring transparent and fully reproducible end-to-end data handling.

We again thank the Reviewers for their detailed and constructive feedback, which substantially strengthened the quality, reliability, and clarity of the manuscript.

---

### Meta-Review · Area_Chair_Z3V1 · 2025-12-18

**Summary:**

The reviewers unanimously recognize the core contributions of this paper: it establishes the first benchmark for aptamer–small molecule interactions, systematically compares shallow and deep learning methods under different data splitting protocols, and identifies ligand representation as the main bottleneck in the task.

However, several major concerns were also raised:

1. Data quality: The current set of small molecules is limited in size and coverage, which may introduce class imbalance. Additionally, the use of synthetically generated negative samples could introduce label noise.

2. Experimental comparisons and evaluation: The reasons behind the underperformance of current deep learning methods relative to shallow models remain unclear. Comparisons should be extended to include interaction-aware deep models, structure-aware methods, and physics-based approaches. More in-depth analysis such as identifying which chemical features are predictive of binding is also recommended.

3. Main insight: The claim that molecule representation is the bottleneck requires further justification. What specific type of representation is needed for this task should be clarified.

4. Motivation: The reported strong performance with only around 2,000 samples seems at odds with the inherent complexity and vast search space of the problem, warranting further explanation.

5. Data accessibility: To serve as a sustainable community benchmark, the data should be made publicly available in a standardized format (e.g., on HuggingFace Datasets) and maintained with versioned updates.

**Reviewer Concerns:**

1. Data quality: Partially addressed. The authors have augmented the dataset with additional molecules and negative samples from existing literature. However, the coverage and representativeness of these additions remain unclear. More detailed statistical analysis, beyond simply reporting counts, is needed to fully assess data quality and balance.

2. Experimental comparisons: Partially addressed. While the authors have introduced baselines such as cross-attention and UniMol to represent interaction-aware and structure-aware methods, physics-based models remain absent from the comparison.

3. Main insight (molecular representation bottleneck): Not addressed. Despite the availability of advanced molecular representation techniques, including graph neural networks, transformers, multimodal approaches, and pre-trained models, the authors have not conducted a thorough investigation into why molecular representation emerges as the bottleneck. A deeper analysis is needed to determine whether the limitation stems from molecular size, representation type, or other factors. Given that this is a key insight of the paper, further validation is essential.

4. Motivation: Addressed. The authors have agreed to revise the wording to better illustrate the challenges of this task.

5. Data accessibility: Addressed. The authors have committed to releasing the dataset on HuggingFace, which will enhance reproducibility and support ongoing community benchmarking.

**Reviewer Scores:**

I think some reviewers may increase their score, e.g. from 2 to 4. However, since there remain some serious issues unaddressed, I don't think the reviewers would change their opinion from reject to accept.

---

### Decision · Program_Chairs · 2026-01-26

Reject